# The Comprehensive Roles of ATRANORIN, A Secondary Metabolite from the Antarctic Lichen *Stereocaulon caespitosum*, in HCC Tumorigenesis

**DOI:** 10.3390/molecules24071414

**Published:** 2019-04-10

**Authors:** Young-Jun Jeon, Sanghee Kim, Ji Hee Kim, Ui Joung Youn, Sung-Suk Suh

**Affiliations:** 1Comprehensive Cancer Center, The Ohio State University, Columbus, OH 43210, USA; jeon.81@osu.edu; 2Division of Polar Life Sciences, Korea Polar Research Institute, Incheon 21990, Korea; sangheekim@kopri.re.kr (S.K.); jhalgae@kopri.re.kr (J.H.K.); ujyoun@kopri.re.kr (U.J.Y.); 3Department of Polar Sciences, University of Science and Technology, Incheon 21990, Korea; 4Department of Bioscience, Mokpo National University, Muan 58554, Korea

**Keywords:** hepatocellular carcinoma (HCC), lichen, atranorin, cell cycle, cell death

## Abstract

Hepatocellular carcinoma (HCC) is one of the most deadly genetic diseases, but surprisingly chemotherapeutic approaches against HCC are only limited to a few targets. In particular, considering the difficulty of a chemotherapeutic drug development in terms of cost and time enforces searching for surrogates to minimize effort and maximize efficiency in anti-cancer therapy. In spite of the report that approximately one thousand lichen-derived metabolites have been isolated, the knowledge about their functions and consequences in cancer development is relatively limited. Moreover, one of the major second metabolites from lichens, Atranorin has never been studied in HCC. Regarding this, we comprehensively analyze the effect of Atranorin by employing representative HCC cell lines and experimental approaches. Cell proliferation and cell cycle analysis using the compound consistently show the inhibitory effects of Atranorin. Moreover, cell death determination using Annexin-V and (Propidium Iodide) PI staining suggests that it induces cell death through necrosis. Lastly, the metastatic potential of HCC cell lines is significantly inhibited by the drug. Taken these together, we claim a novel functional finding that Atranorin comprehensively suppresses HCC tumorigenesis and metastatic potential, which could provide an important basis for anti-cancer therapeutics.

## 1. Introduction

In spite of explosively advancing technology regarding medical genetics, anti-cancer therapeutics remain challenging because of the complexity of tumorigenesis and cancer metastasis. In particular, hepatocellular carcinoma (HCC) is hepatic primary cancer and the sixth- ranked cause of cancer-related deaths worldwide. HCC is often caused by genetic mutations induced by chronic haptic inflammation, viral infection and other environmental factors, and is often incurable when diagnosed at the late stage [1,2,3]. Regardless of a tremendous effort to cure this cancer, an EGFR inhibitor, sorafinib, has been only applied to patients whose effect is also limited in patients with a specific genetic background such as mutations in receptor tyrosine kinases in chemotherapy [4]. Therefore, other chemotherapeutic agents should be investigated and characterized. 

To fully develop a chemotherapeutic drug for cancer to be marketed from in vitro bench work requires at least half a million US dollars, whose cost can reach around 3 billion US dollars in other reports [5,6,7]. Moreover, 10 representative chemotherapeutic drugs against cancers required around 10 years [5]. These are generally considered huge hindrances in anti-cancer therapies and consequently acknowledge the need to investigate other surrogates such as natural derivatives from other organisms [8,9]. 

Lichens are classified as a symbiotic organism generally composed of a fungal partner and a photosynthetic organism such as algae or cyanobacteria. In the symbiosis, it has been known that Lichens often stimulate fungi to produce secondary metabolites having important biological roles from inducing cell division of Lichens to protecting from microbial infection. Currently around 30,000 different species of Lichens have been observed worldwide, and their derivatives are often the source for bioactive substances such as antibiotics [10,11]. Although Lichens and their secondary metabolites have been promising anti-cancer reagents as a natural compound-derived medicine, the effect of those products in cancer is not very conclusive due to a lack of evidence. Moreover, it is a kind of surprise that only a few groups of the lichen family have been reported, although around one thousand secondary metabolites from Lichens have been identified (Figure 1) [11,12,13].

Atranorin, having a depside structure, is one of the major secondary metabolites from Lichens, and exhibits a versatile biological roles, making it an attractive molecule for anti-cancer therapy. It has been widely known that oxidative stress causes several aspects of cellular responses, resulting in the induction of DNA damage response, unfolded protein responses, autophagy and other metabolic stresses, closely related to cancer progression and metastasis [14,15,16,17]. Therefore, the hypothesis regarding tumor suppressive role in Atranorin sounds reasonable. Indeed, Atranorin has a tumor suppressive activity in several solid tumors and leukemia, which seems to be associated with the bona fide regulatory function in redox status in cancer cells [11,18,19,20]. However, little is known about its activity in HCC. To this end, we collected Lichens from Antarctica and extracted Atranorin followed by analyzing tumor suppressive activity in HCC. Here, we showed that Atranorin suppressed the oncogenic properties of HCC by suppressing cell proliferation, survival, cell cycle progression and cell migration/invasion ability.

## 2. Results and Discussion

To directly analyze the effect of Atranorin in HCC tumorigenesis, we selected representative HCC cell lines. SKHep1 and Huh-7 cells are epithelial carcinomas and are known as metastatic cancer cell lines, whereas SNU-182 is a primary cancer cell line. Like the other cancer types, TP53 mutations are accumulated in advanced stages of HCC as well as localized tumors, which frequently drive HCC progression. Both Huh7 and SNU-182 cells harbor the missense mutant allele of the TP53 gene, mostly loss of function mutation. Moreover, the SKHep1 cells have TP53 the wild type allele, but lack the CDKN2A gene, a potent CDK inhibitor Therefore, we performed the experiment to test the effect of Atranorin on the TP53 wild type and mutant [21,22,23,24,25,26].

### 2.1. Atranorin Inhibits HCC Cell Growth

To directly address the anti-tumoric effect of Atranorin on HCC, we first performed the cell proliferation assay using three representative HCC cell lines. The administration of this natural compound into the cells and subsequent cell proliferation assay showed that Atranorin seems to have a marginal effect in a low dose treatment up to 10 ug/mL concentration. However, the cell growth rate was significantly inhibited under the high dose condition (Figure 2A). Consistent with the previous biochemical assay, the cell growth and morphology were also monitored under a light microscope, proving that the number of cells was consistently reduced in response to Atranorin treatment (Figure 2B). 

### 2.2. Cell Cycle in HCC is Attenuated by Atranorin Treatment

Cell cycle progression is closely related to cell growth, tightly regulated by cyclin-dependent kinases (CDKs). The somatic mutations on the cyclins and CDK genes are frequently observed in cancers. Consequently, the failure of the regulation in cell cycle machinery generally drives cancer progression [14]. To avoid unwanted mitogen messaging caused by mutations on the CDKs and cyclin genes, we tried to select cells harboring wild type alleles of those genes. According to the data set in Cancer Cell Line Encyclopedia (CCLE), no mutation in CDKs and cyclins is found in SK-Hep1, Huh-7 and SNU-182 HCC cell lines. Therefore, we decided to analyze the cell cycle progression with HCC cell lines harboring the wild type in Atranorin-induced cell growth inhibition. We investigated the ability of the cell cycle in SK-Hep1 cells treated with different concentrations of Atranorin (0–40 μg/mL) by analyzing the amount of DNA contents with propidium iodide (PI) staining. As shown in Figure 2, we found that a lower number of the cells treated with Atranorin were significantly stocked in S phase, which consequently increased the number of cells in G2/M phase (Figure 3). These data suggest that Atranorin treatment evoked G2/M phase cell cycle arrest. Furthermore, our observation that the number of cells going through S phase was reduced suggests that Atranorin could induce G1 phase cell cycle arrest, as well as G2-M phase. 

Reactive Oxygen Species (ROS) are quite common in cancer due to an enhanced metabolic rate and cell proliferation. In terms of cancer progression, the ROS functions like a double-edged sword. Low exposure of ROS to cancer cells, for example, enhances the cell cycle like a mitogen whereas acute and high concentrations of subcellular ROS level often suppress cancer progression [27]. Considering that the known activity of a secondary lichen metabolite is a ROS regulator, we, therefore, believe that Atranorin-induced cell cycle regulation is likely to be related to ROS regulation in HCCs [13]. 

### 2.3. Atranorin Enhances Cell Death

Programmed cell death, better known as apoptosis, should be tightly monitored and regulated to maintain cell homeostasis whose failure is often directly connected to several diseases. One of the prominent examples of the consequence of apoptotic resistance is cancer [28]. In addition to the inhibitory effect on cell growth, we also observed morphological changes (Figure 2B), which motivated us to analyze the other anti-tumoric effects. Therefore, we hypothesized that Atranorin is likely to induce stress signals such as inducing cell death. To directly address this hypothesis, we performed FACS analysis by employing an apoptotic and a necrotic marker. The cells were subcultured in 6-well plates and subsequently were treated with Atranorin. After 24 h, the cells were harvested and stained by Annexin-V5 to measure the apoptotic rate and by PI for late apoptosis and/or necrosis. As a result, we found that the cells treated with Atranorin underwent severe cell death in SNU-182 and Huh-7 cells compared to non-treated control cells (Figure 4A,B). However, we did not observe significant change in SK-Hep1 cells (data not shown). Additionally, the summarized data indicated that there were more dying cells double positive for annexin-V and PI, or single positive for PI rather than those with annexin-V positive only, suggesting that the necrotic cell death mediated by cell rupture or lysis could be a major source of cell death evoked by Atranorin rather than typical apoptotic mechanism. This is an interesting observation that a typical second Lichen metabolite is generally known as an anti-inflammatory molecule, which could be opposite from our observation [29]. However, the action of Atranorin as either a pro- or anti-oxidant molecule is dependent on the sorts of free radicals in the cell, suggesting that Atranorin could play a pro-oxidant role in HCCs [13]. Also, it cannot be ruled out that Atranorin-induced cell death is an early event, which consequently increased the number of cells going through necrotic-like cell death at the time point when we observed. However, it should be interesting to determine how Atranorin regulates ER stress, hypoxic response and/or autophagy, which all consequently represent necrotic cell death. 

### 2.4. Atranorin Enhances Metastatic Potential of HCC

Cancer metastasis is characterized by a multistep inefficient process, but its consequence is often deadly. As a consequence of the changed cell-cell interaction and its microenvironmental factors, cancer cells could gain the abilities of migration and invasion often represented for cancer metastasis [1,30]. Regarding this process, genetic and epigenetic regulators have been suggested to suppress this mechanism, but no applicable factor has been identified well. Our previous observation that Atranorin exhibited a tumor suppressive effect on cell cycle progression in HCC cells led us to analyze the druggable potential as an anti-metastatic molecule. 

The Sk-Hep1 and Huh-7 cells were plated in a 100-mm dish, and wound-healing ability was subsequently addressed by a scratch assay. As a result, we found that Atranorin treatment significantly inhibited the healing ability (Figure 5A,B). Interestingly, we previously showed Atranorin did not show any inhibitory effect in cell proliferation at a low dose (Figure 2) suppressing cell wound-healing ability. These observations suggest that wound-healing suppression by Atranorin is likely to be mediated by suppressing cell migration. Therefore, we investigated whether the natural drug could also suppress cell migration and invasion by employing specific chambers. As shown in Figure 6, the SK-Hep1 and Huh-7 cells were incubated with Atranorin for 24 h followed by placing a migration or an invasion chamber for an additional 24 h. As a result, the migratory and invasive effects of the cells were suppressed upon Atranorin treatment (Figure 6A,B). Taken together, here we are claiming that a Lichen-derived secondary metabolite, Atranorin suppressed the oncogenic potential of HCC by suppressing cell proliferation, migration and invasion capacities, which provides insight into anti-cancer therapeutics against HCC by providing functional evidence. 

## 3. Materials and Methods

### 3.1. Lichen Material

The lichen, *Stereocaulon caespitosum*, was collected in January 2017 from King George Island, Antarctic Peninsula, Antarctica, (62°12′53.69″ S; 58°55′23.87″ W), and identified by Dr. Ji Hee Kim and Miss Jae Eun So. A voucher specimen (no. Ant-061) was deposited at the Natural Product Chemistry Laboratory of the Korea Polar Research Institute (Incheon, South Korea).

### 3.2. Extraction and Isolation

The air-dried and powdered lichen, *S. caespitosum* (100 g), was extracted by maceration in methanol (MeOH) (3 × 0.5 L) at room temperature. The solvent was concentrated in vacuo to yield 5 g of a crude extract, which was then suspended in distilled water (0.2 L) and extracted successively with *n*-hexane (2 × 0.5 L), ethyl acetate (EtOAc) (2 × 0.5 L), and *n*-butanol (2 × 0.5 L). The EtOAc extracts (2.3 g) were separated by column chromatography (CC) over a C18 gel column and eluted with MeOH:H_2_O, (10:90 to 100% MeOH) to obtain 18 subfractions (ER1 to ER18). Atranorin (**2**, 10 mg) was obtained as a white amorphous powder by recrystallization in 100% chloroform (CHCl_3_) from subfraction ER12 (50 mg), which was further purified by semi-preparative HPLC (Young Lin, South Korea) on an Alltech reversed-phase YMC-Pak C-18 column (10 μm, 20 × 250 mm) with a flow rate of 2 mL/min, using MeOH:H_2_O solvent mixtures (from 20:80 to 100% MeOH).

### 3.3. Cell Culture and Maintenance 

The HCC cell lines used in the current study were obtained from the Korean Cell Line Bank (KCLB; cellbank.snu.ac.kr) and cultured in RPMI 1640 supplemented with 10% Fetal Bovine Serum (FBS). Periodically, the cells were monitored using specific kits to detect bacterial and/or mycoplasma contamination (Invivogen, SD, USA) and ThermoFisher Scientific, Waltham, MA, USA).

### 3.4. Cell Proliferation Assay

Cell viability was determined by MTS assay as per the manufacturer’s protocol (Promega). In brief, the cells were subcultured in 96-well plates and the desired drug such as Atranorin was treated as described in the result and/or figure legends. After that, 10 uL of MTS solution was added into the plate and additionally incubate in cell culture incubator for 10–60 min followed by measuring observance at 490 nm.

### 3.5. Cell Cycle Assay

SK-Hep1 cells were seeded in six-well plates at a density of 2.5 × 10^5^ cells/mL, grown for 24 h, and treated with Atranorin at the indicated concentrations. After 24 h of treatment, the cells were washed with phosphate-buffered saline (PBS), fixed with 70% ethanol and stored at −20 °C overnight. For analysis, cells were washed, resuspended in PBS, and incubated with 5 μL of RNase A (10 mg/mL) and 5 μL of PI (50 μg/mL) at 37 °C for 30 min. After incubation, they were analyzed using a flow cytometer (Beckman Coulter Inc., Brea, CA, USA)—20,000 ungated events were analyzed.

### 3.6. Cell Death Determination Using FACS

The cells treated with Atranorin were prepared for staining with an apoptosis detection kit (TheromoFisher Scientific). The desired cells were stained using Annexin V conjugated with FITC and Propidium Iodide with PE as recommended in the manufacturer’s protocol. Subsequently, the cell death was determined by FACScalibur (Beckton Dickinson, NJ, USA). Overall, numbers of cells having positive signals in each panel were summarized in the figure. 

### 3.7. Wound-Healing Assay

The indicated cells were subcultured in a 6-well plate for 24 h and the wound was addressed by gently scraping the surface with a sterile p200 pipette tip. After 24 h, images of migrated cells in the wound area were captured using a digital camera. All scratch assays were performed in triplicate.

### 3.8. Migration and Invasion Assay

The overall experimental procedures were conducted following the manufacturer’s protocol (Calbiochem). Briefly, the cells were placed into 6-well plates and were subsequently placed in the reduced serum medium (1% FBS) for 16 h. After that, the cells were treated with Atranorin as shown in the figure followed by placement in a migration or invasion chamber. Additionally, the complete media containing 10% FBS was added into lower chamber, which could be used as a chemoattractant. After 24–48 h, the migrated and invasive cells were stained using a Cell Stain Solution (400 μL) and were photographed.

### 3.9. Statistical Analysis

Data are expressed as the mean ± the standard errors of the mean (SEM). Statistical difference was evaluated with Student’s *t* test. *P* values lower than 0.05 were considered to represent significant differences.

## 4. Conclusions

In this study, we claim that Atranorin, a secondary metabolite derived from Antarctic *Stereocaulon caespitosum* has comprehensive tumor suppressive roles in hepatocellular carcinoma (HCC) through regulation of the cell cycle, cell death, and metastatic potential. To our knowledge, this is the first report to analyze the role of Atranorin in HCC tumorigenesis, although how it affects HCC progression should be further investigated with in vivo validation such as tumor xenograft models. Also, it could be an interesting subject to study Atranorin as a modifier of intracellular redox potential to suppress HCC progression. Collectively, our current observations indicate that Atranorin plays an important role in HCC progression, providing potentially applicable insight by having tremendous benefits as a natural compound from lichens.

## Figures and Tables

**Figure 1 molecules-24-01414-f001:**
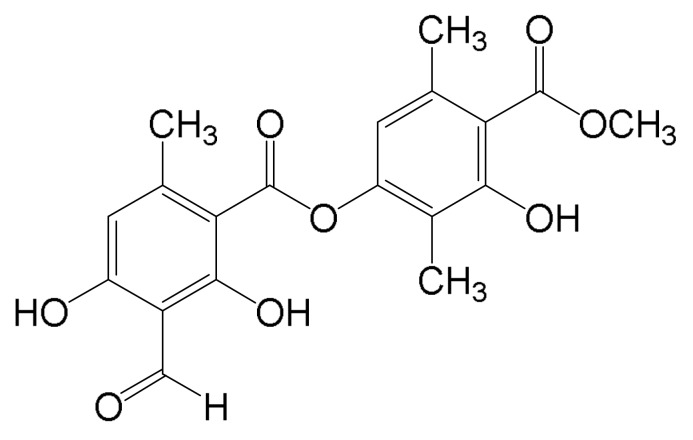
Chemical structure of Atranorin (molecular weight [Mw], 374.34).

**Figure 2 molecules-24-01414-f002:**
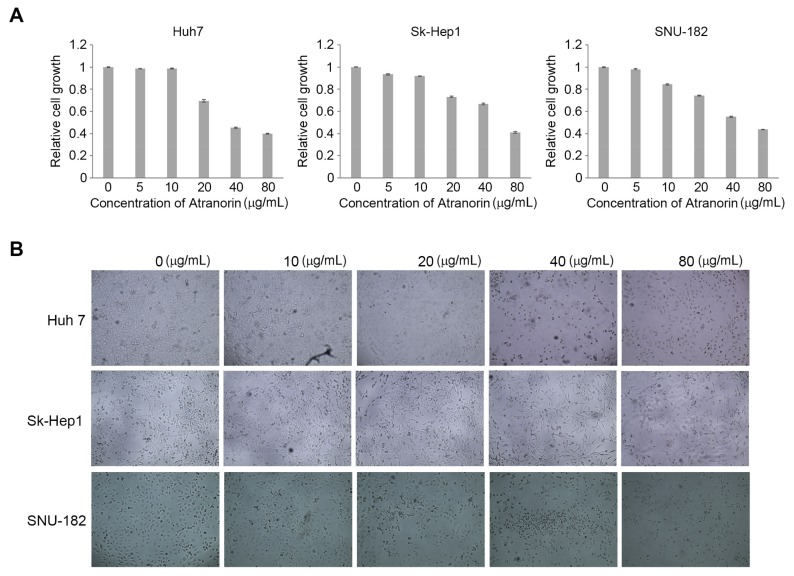
Atranorin inhibits cell growth in a dose-dependent manner. (**A**) MTT assay using HCC cell lines. The indicated cell lines were plated into 96 well pate and Atranorin was treated as shown in the figure for 24 h. Subsequently, the cell growth was measured by a standard MTT assay and the values were normalized by those of the non-treatment group. Bars represent ±SEM, and p-values were obtained by paired Student’s *t*-test (*n* = 3, *p* < 0.05). (**B**) Microscopic images of the HCC cells treated with Atranorin. The indicated cells were treated with Atranorin for 24 h followed by monitoring the changed number of cells and morphology under light microscope.

**Figure 3 molecules-24-01414-f003:**
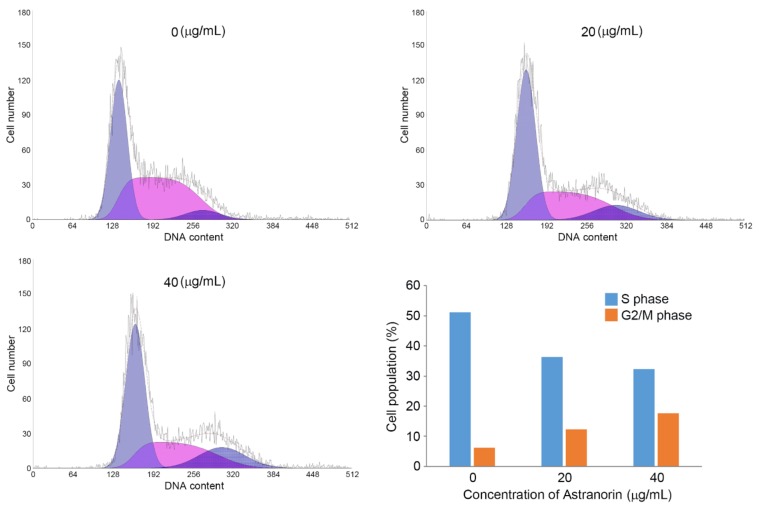
The inhibitory effect on the cell cycle by Atranorin in HCC. SK-Hep1 cells were treated with Atranorin and subsequently the samples were prepared for cell cycle analysis by adding propidium iodide. Each cell cycle step was determined by DNA contents on the *x*-axis and number of events on the *y*-axis. The summarized result is shown in the right bottom panel.

**Figure 4 molecules-24-01414-f004:**
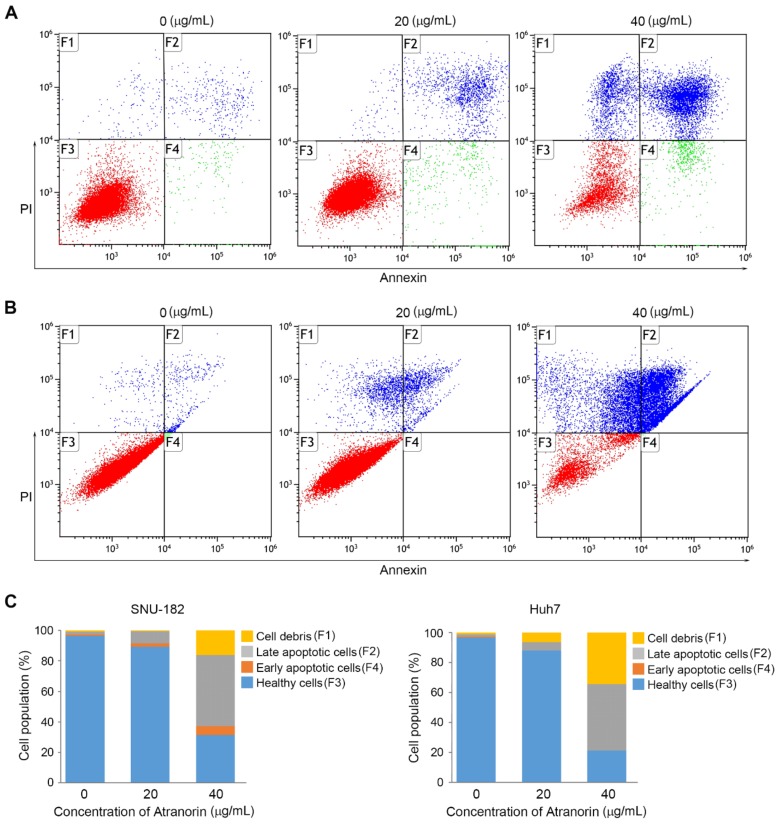
Atranorin treatment sensitizing HCCs. (**A**,**B**) the analysis of apoptotic events in the SNU-182 (**A**) or Huh-7 (**B**) cells. The cells were incubated with Atranorin for 24 h as shown in the figure. Subsequently, the samples were treated with annexin-V and Propidium Iodide (PI) followed by counting the positive events under FACS. The quadruplicated plot representing early apoptosis (annexin-V+ and PI-), late apoptosis and/or necrosis (annexin-V+/- and PI+), respectively. (**C**) The summarized analysis of the apoptotic events enforced by Atranorin in SNU-182 (left panel) and Huh-7 (right panel).

**Figure 5 molecules-24-01414-f005:**
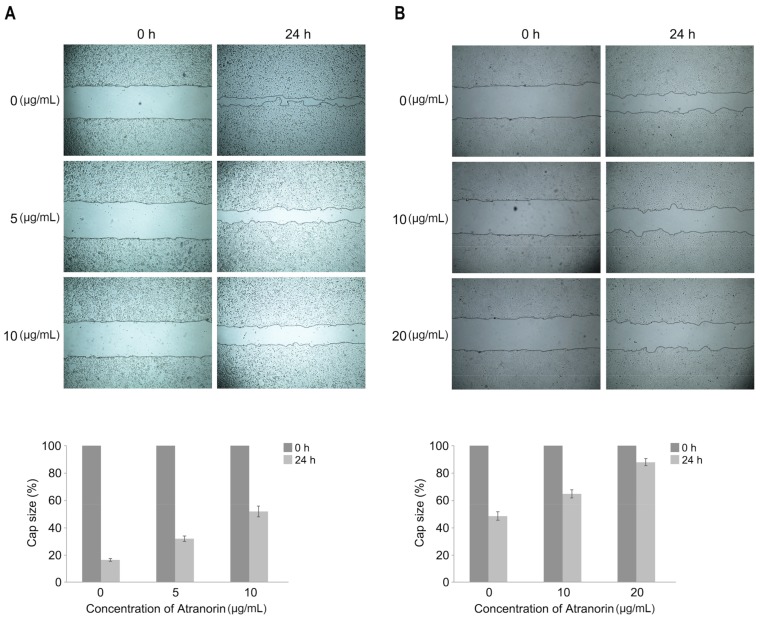
Suppressed wound-healing ability in HCCs by Atranorin. The SK-Hep1 (**A**) and Huh-7 (B cells were placed into a 6-well plate. After that, the cells were scratched to make a gap by pipette tip followed by monitoring migratory and/or wound-healing ability of the cells under a microscope (**A**,**B**, upper panels). The gap-filling capacity was determined by measuring the distance between the gabs using image J software. The overall effect was summarized in the bottom panel of each figure. Bars represent ±SEM, and *p*-values were calculated by a paired Student’s *t*-test (*n* = 3, *p* < 0.05).

**Figure 6 molecules-24-01414-f006:**
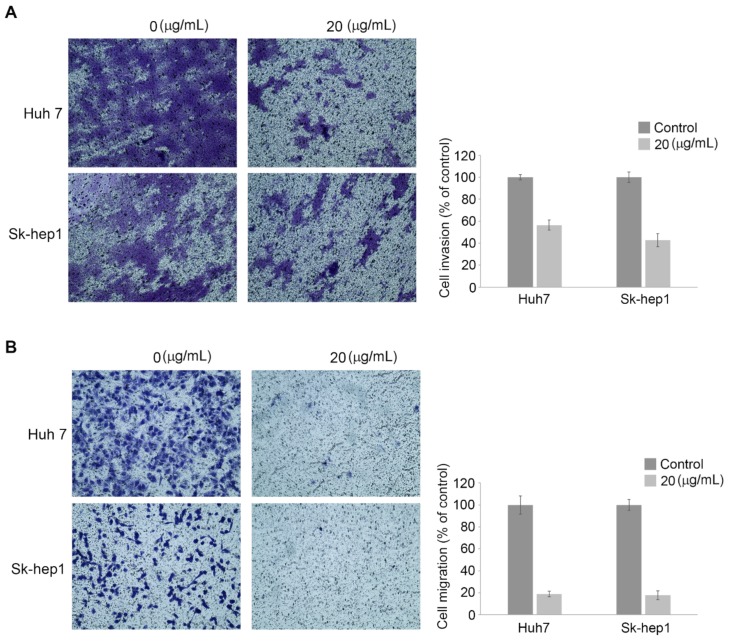
Atranorin treatment suppressing metastatic potential of HCC by suppressing invasion and migration abilities. The Huh-7 or SK-Hep1 cells were incubated with DMSO or Atranorin for 24 h. Subsequently, 5 × 10^5^/mL of the cells were plated into upper chamber in 100 uL medium supplemented with 0.1% FBS. The normal medium containing 10% FBS was added into the bottom chamber used as a chemoattractant. After 24 h, the migratory and invasive cells were monitored. The data on the right side of each figure are summarized. Bars indicate ±SEM, and p-values were calculated by paired Student’s *t*-test (*n* = 3, *p* < 0.05).

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
