# Peer review of "The Comprehensive Roles of ATRANORIN, A Secondary Metabolite from the Antarctic Lichen *Stereocaulon caespitosum*, in HCC Tumorigenesis"

_molecules, 2019, doi:10.3390/molecules24071414_

Round 1
Reviewer 1 Report
I suggest the Authors to improve the introduction section by adding more information on secondary metabolites (see Molecules 23(2), 463, 1-11, 2018; Molecules 23(2), 461, 1-14, 2018; Food and Chemical Toxicology, 119, 189-198, 2018; Industrial Crops and Products, 130, 9-15, 2019).
Furthermore i suggest to improve the English language and particularly, to improve the characterization section by adding instrumental analysis of the extract (e.g. HPLC) in order to check at least the major components
Author Response
Comments and Suggestions for Authors
Response: We appreciate the time and efforts of the reviewer in consideration of the original manuscript and believe that the comments have helped us to improve the manuscript. All the corresponding changes in response to the reviewer’s comments are marked in red.
Comment 1: I suggest the Authors to improve the introduction section by adding more information on secondary metabolites (see Molecules 23(2), 463, 1-11, 2018; Molecules 23(2), 461, 1-14, 2018; Food and Chemical Toxicology, 119, 189-198, 2018; Industrial Crops and Products, 130, 9-15, 2019).
Response: First, we greatly appreciate your comments, critical to improve overall quality of this manuscript. As recommended from the reviewer, we provided more information about the secondary metabolites from Lichens.
Comment 2: Furthermore I suggest to improve the English language and particularly, to improve the characterization section by adding instrumental analysis of the extract (e.g. HPLC) in order to check at least the major components.
Response: Thank you for your suggestion. As the reviewer pointed out, we carefully reviewed the manuscript. In particular, we reorganized some parts of the introduction section with more information, and made some changes for grammatical errors. In response to reviewer’s recommendation, we also added more information for HPLC analysis of the extract to improve characterization section in Material and Methods section.

Reviewer 2 Report
The work by Y. Jeon et al has been well performed and will be of interest for the scientific community. Please find below some changes and corrections to be done:
- In second page second paragraph says “as a fork medicine” but this is not a correct term, please use a correct term.
- In second page second paragraph says “although around one thousand secondary metabolites from Lichens have identified [10].” but should say “although around one thousand secondary metabolites from Lichens have been identified [10].”
- In second page second paragraph says “depside structure with polyphenolic carbon ring and exhibits a versatile biological role, which has been particularly an attractive molecule for anti-cancer therapy” but should say “depside structure and exhibits a versatile biological role, what makes it an attractive molecule for anti-cancer therapy”
- In second page second paragraph says “which has been being confirmed” but should say “which has been confirmed”
- In Figure 2 reading says “the indicted cells” but should say “the indicated cells”
- Term “Atranorin” is written with initial capital letter but sometimes along the text it is not, it should be consistent.
- In Figure 3 reading says “propium iodide” but should say “propidium iodide”
- In page 4 says “Apoptosis” but should say “apoptosis”
- In page 4 says “PI” but it should be indicated the meaning first time used “propinium iodide (PI)”
- In page 4 says “Annexin-5” and “Annexin-V5” but should say “ annexin-V”
- In page 4 says “cells treated with antranorin were undergone” but should say “cells treated with antranorin undergone”
- In references 8 and 13, second page is missing
Author Response
Comments and Suggestions for Authors
The work by Y. Jeon et al has been well performed and will be of interest for the scientific community. Please find below some changes and corrections to be done:
Response: We appreciate the time and efforts of the reviewer in consideration of the original manuscript to improve the manuscript. All changes in the text (additions and deletions over the previous submission) are marked in red.
Comment 1: In second page second paragraph says “as a fork medicine” but this is not a correct term, please use a correct term.
Response: As the reviewer pointed out, the term “as a fork medicine” is not technically right term. Thus, we changed this term into a correct term. “As a natural compound-derived medicine”
Comment 2: In second page second paragraph says “although around one thousand secondary metabolites from Lichens have identified [10].” but should say “although around one thousand secondary metabolites from Lichens have been identified [10].”
Response: We fixed it.
Comment 3: In second page second paragraph says “depside structure with polyphenolic carbon ring and exhibits a versatile biological role, which has been particularly an attractive molecule for anti-cancer therapy” but should say “depside structure and exhibits a versatile biological role, what makes it an attractive molecule for anti-cancer therapy”
Response: We replaced it as recommended.
Comment 4: In second page second paragraph says “which has been being confirmed” but should say “which has been confirmed”
Response: We fixed it.
Comment 5: In Figure 2 reading says “the indicted cells” but should say “the indicated cells”
Response: We fixed the typo.
Comment 6: Term “Atranorin” is written with initial capital letter but sometimes along the text it is not, it should be consistent.
Response: As the reviewer pointed out, we replaced all the term with initial capital letter.
Comment 7: In Figure 3 reading says “propium iodide” but should say “propidium iodide”
Response: We fixed it.
Comment 8: In page 4 says “Apoptosis” but should say “apoptosis”
Response: We fixed it.
Comment 9: In page 4 says “PI” but it should be indicated the meaning first time used “propinium iodide (PI)”
Response: We changed it as recommended.
Comment 10: In page 4 says “Annexin-5” and “Annexin-V5” but should say “ annexin-V”
Response: In addition to the wrong terms in page 4, we found the term used in different page, and we replaced all into “annexin-V” as recommended.
Comment 11: In page 4 says “cells treated with antranorin were undergone” but should say “cells treated with antranorin undergone”
Response: We changed it.
Comment 12: In references 8 and 13, second page is missing
Response: We are not sure what exact meaning of this comment. It would be greatly appreciated if we can get more detailed comment about this. Thanks very much.

Reviewer 3 Report
Authors describe the inhibitory role of atranorin on hepaocellular carcinoma. This is based on inhibition of cell growth, metastatic potential and induction of necrosis in HCC cell lines and metastatic models.
General comment: please provide more information on the used cell lines. What are their differences, in which way they represent HCC patients and carcinogenesis paths including viral carcinogenesis?
Not all analysis methods have been completed on all cell lines. Please provide infomation, which method (as cell cycle analyis and cell death FACS analysis) has been done on which cell line, and why athors find these cells as representative. If other cell lines are also used, and even the results were not consistent, please also mention these results in the Results text.
Results
Figure 2. The middle panel is rather dark. Please provide better visible images.
Paragraph 2-2. Please mention if any of the used HCC cell lines have CDK or cyclin mutations.
Paragraph 2-2. Do the authors have measured evidence on ROS changes in untreated and atranorin treated conditions?
Please thoroughly rewrite the paragraph 2-4. The scratch would healing assay is not a metastatic model, and without the suppression of cell division by i. e. Mitomycin C, the scatched gap can be also filled by cell division. Please only mention that the scratch assay was done, and do not interpret this result as evidence for suppression of metastasis, the migration chamber results and interpretation are fine.
Please improve the quality of the images of Figure 5.
Author Response
Comments and Suggestions for Authors
Authors describe the inhibitory role of atranorin on hepaocellular carcinoma. This is based on inhibition of cell growth, metastatic potential and induction of necrosis in HCC cell lines and metastatic models.
Response: We appreciate the comments of the reviewer in consideration of the original manuscript, which helped us to improve the manuscript. All changes in the text (additions and deletions over the previous submission) are marked in red.
General comment: please provide more information on the used cell lines. What are their differences, in which way they represent HCC patients and carcinogenesis paths including viral carcinogenesis?
Response: Thank you for your comment. As the reviewer pointed out, we provided more information on the three cell lines in the early result section. Those all cell lines used in the current study are epithelial carcinomas. Sk-Hep1 and Huh-7 cells are known as a metastasized cancer cell lines whereas SNU-182 was isolated from primary hepatocellular carcinoma. One of the most important criteria to select these cell lines in the current study was the status of TP53 mutation known as the most dysregulated and mutated gene in most types of cancer patients. In particular, both Huh 7 and SNU-182 expresses missense mutant allele of TP53 gene, mostly likely being loss of function mutations. In particular, a potent CDK inhibitor, CDKN2A is known to be homozygosly deleted in Sk-Hep1 cells harboring wild type allele of TP53 gene [1-6].
Therefore, the Atranorin-mediated tumor suppressive effects could function on advance stage of HCC having metastatic characteristics with hyper-somatic mutation on TP53 gene. Regarding this comment, we added more information about the cell lines in the result section.
Comment: Not all analysis methods have been completed on all cell lines. Please provide information, which method (as cell cycle analysis and cell death FACS analysis) has been done on which cell line, and why authors find these cells as representative. If other cell lines are also used, and even the results were not consistent, please also mention these results in the Results text.
Response: We provided information on the cell lines used in each analysis in the corresponding figure legends or “Material and Method” section. In general, although we only showed representative result in some experiment, actually performed the experiments with three cell lines, and got similar effects, otherwise we described in the result section. In particular, we only observed the significant cell death in both Huh7 and SNU-182 cells, but not in SK-Hep1 cells. However, Atranorin-dependent cell cycle arrests were observed in all three cell lines that we tested. At this point, we do not have any evidence how Atranorin-induced cell death is context dependent, which should be further addressed. Regarding this comment, we added this result in the result section.
Comment: Figure 2. The middle panel is rather dark. Please provide better visible images.
Response: In response to the reviewer’s recommendation, we replaced the middle panel into brighter ones.
Comment: Paragraph 2-2. Please mention if any of the used HCC cell lines have CDK or cyclin mutations.
Response: We especially thank the reviewer for this comment. To analyze the cell cycle in Atranorin-treated cell, we believe any mutation on the cell cycle-related genes could provide confusing message rather than from the Atranorin. Thus, we initially tried to find the cells harboring wild type alleles in CDKs and cyclin genes, and decided to use those cell lines based on the observation that no mutation has not been identified in those genes as shown in CCLE (Cancer Cell Line Encyclopedia) project. Additionally, SKHep1 harboring homozygous mutation on CDKN2A, an inhibitor of CDKs, was used for monitoring cell cycle regulated by Atranorin in this study [1]. Now we reflected this response to our manuscript in the section 2-2.
Comment: Paragraph 2-2. Do the authors have measured evidence on ROS changes in untreated and atranorin treated conditions?
Response: Thank you for your comment. At this point, we do not have any experimental evidence on Atranorin-induced ROS change, which could be very crucial to characterize how Atranorin regulates cancer progression in HCC. As shown in initial manuscript, Atranorin is, however, known to regulate intracellular redox potentials, which depends on different intracellular radicals as an anti- or pro-oxidant molecule [7]. Therefore, it could be interesting follow up study to characterize how the activity of the Atranorin on intracellular free radicals is related with Atranorin-regulated HCC progression as pointed out by the reviewer.
Comment: Please thoroughly rewrite the paragraph 2-4. The scratch would healing assay is not a metastatic model, and without the suppression of cell division by i. e. Mitomycin C, the scatched gap can be also filled by cell division. Please only mention that the scratch assay was done, and do not interpret this result as evidence for suppression of metastasis, the migration chamber results and interpretation are fine.
Response: As the reviewer indicated, we totally agree that the scratch is only to characterize migration and proliferation ability of a cell. In case of our setting, we used 5 or 10 ug/ml of Atranorin for the scratch assay, and did not observe significant inhibition of cell proliferation in Huh 7 cell and only marginal suppression in Sk-Hep1 cell (Figure 2A). However, our scratch assay showed significant suppression with Atranorin treatment using the same concentration with Figure 2A (5 or 10 ug/ml). Therefore, the Atranorin-dependent suppressive activity is likely to be mediated by suppressing cell migratory ability. Nevertheless, we also agree that only these in vitro studies like migration and invasion chambers are not presentative experiment to test HCC metastasis as well as the scratch assay. Thus, we removed the term “metastatic potential” as recommended by reviewer. Moreover, we are willing to change any over-interpreted results into correct ones if the reviewer has more suggestion for this issue.
Comment: Please improve the quality of the images of Figure 5.
Response: Thank for your comment. We tried to get better visible image, but unfortunately have a limited quality of the image due to technical limitation in the microscope for the original images. However, we tried to show this with the better images by changing contrast.
1. Mavrakis KJ, McDonald ER, 3rd, Schlabach MR, Billy E, Hoffman GR, et al. (2016) Disordered methionine metabolism in MTAP/CDKN2A-deleted cancers leads to dependence on PRMT5. Science 351: 1208-1213.
2. Sainz B, Jr., TenCate V, Uprichard SL (2009) Three-dimensional Huh7 cell culture system for the study of Hepatitis C virus infection. Virol J 6: 103.
3. Giacomelli AO, Yang X, Lintner RE, McFarland JM, Duby M, et al. (2018) Mutational processes shape the landscape of TP53 mutations in human cancer. Nat Genet 50: 1381-1387.
4. Park JG, Lee JH, Kang MS, Park KJ, Jeon YM, et al. (1995) Characterization of cell lines established from human hepatocellular carcinoma. Int J Cancer 62: 276-282.
5. Kang MS, Lee HJ, Lee JH, Ku JL, Lee KP, et al. (1996) Mutation of p53 gene in hepatocellular carcinoma cell lines with HBX DNA. Int J Cancer 67: 898-902.
6. Hsu IC, Tokiwa T, Bennett W, Metcalf RA, Welsh JA, et al. (1993) p53 gene mutation and integrated hepatitis B viral DNA sequences in human liver cancer cell lines. Carcinogenesis 14: 987-992.
7. Melo MG, dos Santos JP, Serafini MR, Caregnato FF, Pasquali MA, et al. (2011) Redox properties and cytoprotective actions of atranorin, a lichen secondary metabolite. Toxicol In Vitro 25: 462-468.

Round 2
Reviewer 1 Report
The paper can be accepted
Reviewer 3 Report
Authors responded to all comments and completed the required revision.